# Auslan-Daily: Australian Sign Language Translation for Daily Communication and News

**Xin Shen**[1]    **Shaozu Yuan**[2]    **Hongwei Sheng**[1]    **Heming Du**[1]    **Xin Yu**[1*]

[1]The University of Queensland
[2]JD AI, Beijing, China
x.shen3@uqconnect.edu.au

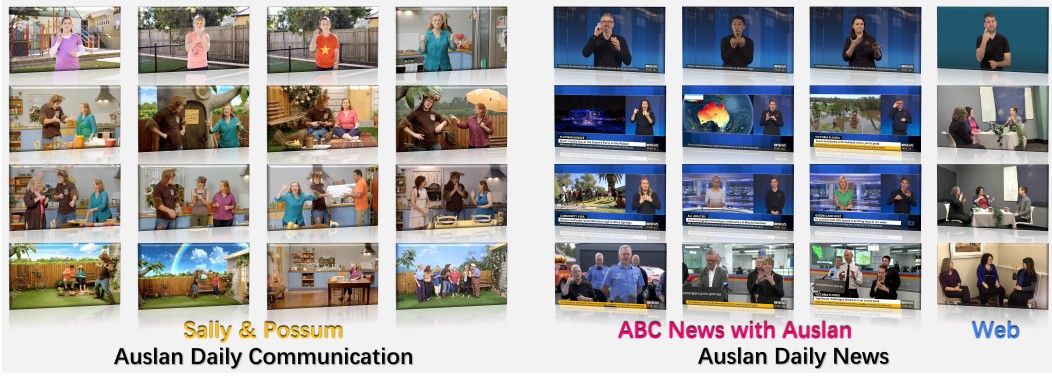

Figure 1: Diversity of our curated Auslan-Daily dataset. Auslan-Daily involves various topics, signers and diverse environments. In particular, there are multiple persons in a scene and signers appear in different areas of the scene.

## Abstract

Sign language translation (SLT) aims to convert a continuous sign language video clip into a spoken language. Considering different geographic regions generally have their own native sign languages, it is valuable to establish corresponding SLT datasets to support related communication and research. Auslan, as a sign language specific to Australia, still lacks a dedicated large-scale dataset for SLT. To fill this gap, we curate an Australian Sign Language translation dataset, dubbed Auslan-Daily, which is collected from the Auslan educational TV series and Auslan TV programs. The former involves daily communications among multiple signers in the wild, while the latter comprises sign language videos for up-to-date news, weather forecasts, and documentaries. In particular, Auslan-Daily has two main features: (1) the topics are diverse and signed by multiple signers, and (2) the scenes in our dataset are more complex, *e.g.*, captured in various environments, gesture interference during multi-signers' interactions and various camera positions. With a collection of more than 45 hours of high-quality Auslan video materials, we invite Auslan experts to align different fine-grained visual and language pairs, including video ↔ fingerspelling, video ↔ gloss, and video ↔ sentence. As a result, Auslan-Daily contains multi-grained annotations that can be utilized to accomplish various fundamental sign language tasks, such as signer detection, sign spotting, fingerspelling detection, isolated sign language recognition, sign language translation and alignment. Moreover, we benchmark results with state-of-the-art models for each task in Auslan-Daily. Experiments indicate that Auslan-Daily is a highly challenging SLT dataset, and we hope this dataset will contribute to the development of Auslan and the advancement of sign languages worldwide in a broader context. All datasets and benchmarks are available at ⚙ Auslan-Daily.

---

*Corresponding author.

37th Conference on Neural Information Processing Systems (NeurIPS 2023) Track on Datasets and Benchmarks.

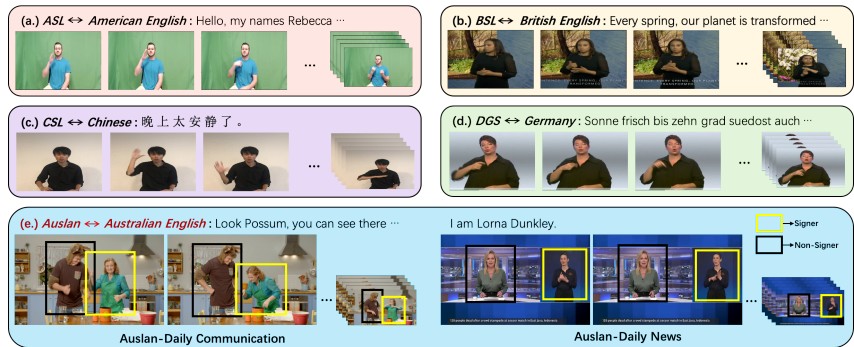

Figure 2: Comparisons of large-scale sign language translation datasets across diverse geographic regions. (a) How2Sign [1] (American), (b) BOBSL [2] (British), (c) CSL-Daily [3] (Chinese), (d) RWTH-PHOENIX-Weather 2014T [4] (Germany), (e) Our proposed **Auslan-Daily** (the yellow and black bounding-boxes indicate the signer and non-signer in the sign video clip).

# 1 Introduction

Sign language (SL) is the primary way for deaf or people with hearing loss to express themselves. Similar to various spoken languages, sign languages have their vocabularies and grammar [5, 6, 7]. More importantly, diverse geographic regions usually have their native sign languages even though these regions share a commonly spoken language, such as America, Australia and the UK. To eliminate the communication barriers between the deaf and hearing communities, sign language translation (SLT) has been proposed to convert signs into spoken languages [1, 2, 4, 3].

With the emerging deep learning techniques and large-scale sign language datasets, SLT has achieved promising progress recently. As shown in Figure 2, researchers from various countries have constructed their sign language datasets and thus thrust SLT in their respective sign languages, such as American sign language (ASL) [1], British sign language (BSL) [2], Chinese sign language (CSL) [3] and Germany sign language (DGS) [4]. However, to the best of our knowledge, there is no publicly available large-scale Auslan dataset for continuous sign translation[2]. Meanwhile, according to the Hearing Care Industry Association[3], as of June 2015, one in six Australians had hearing loss affecting them and this proportion is expected to increase to one in four by 2050. Due to the societal inclusion and the regional nature of sign languages, Australian sign language (Auslan) datasets are inevitably and urgently needed in order to investigate automatic translation.

Moreover, existing sign language corpora either are captured in controlled laboratory environments [1, 3, 8, 9, 10, 11] or only contain a single person (the signer) at a certain position in each video clip [2, 4, 12, 13]. In [14], Núñez-Marcos *et al.* point out that the effectiveness of sign language translation can be compromised in multi-individual situations due to the presence of non-signers. Additionally, Yin *et al.* [15] emphasise that translating sign language in the wild is even more challenging. Thus, the existing datasets lack sufficient diversity and may not fully reflect the complexity in the wild.

In this work, we aim to construct a large-scale Auslan translation dataset in the wild which contains sufficient high-quality Auslan videos and their English transcriptions. We adopt the first Australian educational TV series for deaf children "Sally and Possum"[4], "ABC News with Auslan"[5] and several online domain-specific Auslan corpora[6] as our data source. Firstly, "Sally and Possum" is aired to help deaf children and their parents to learn Auslan, and it covers various topics of daily life, such as drawing pictures with colourful pigments and asking for advice in diverse environments, including indoor and outdoor scenes (Figure 1, left). Secondly, "ABC News with Auslan" starts broadcasting form 2022 and provides the latest news and information from ABC News every week with Auslan interpreted (Figure 1, middle). Lastly, we aggregate additional content-specific Auslan data from

---

[2]Sign videos are segmented and annotated at the sentence level.

[3]https://www.hcia.com.au/resources/HCIA.pdf

[4]https://earlychildhood.qld.gov.au/early-years/sally-possum

[5]https://iview.abc.net.au/show/abc-news-with-auslan

[6]https://www.youtube.com/user/InfoVicdeaf

Table 1: Comparison between Auslan-Daily and existing SLT datasets. PPC.: persons per clip.

| Dataset | SL | Video | Vocab. | #Signer | Source | Background | PPC. | Signer Position | Expert Check |
|---|---|---|---|---|---|---|---|---|---|
| PHOENIX-2014T[4] | DGS | 8K | 3K | 9 | TV | Fixed | 1 | Fixed | ALL |
| SIGNUM[10] | DGS | 33K | 450 | 25 | Lab | Fixed | 1 | Fixed | ALL |
| Content4All[12] | DSGS+VGT | 15k | - | - | TV | Fixed | 1 | Fixed | ALL |
| KETI[9] | KSL | 15K | 0.5K | 14 | Lab | Fixed | 1 | Fixed | ALL |
| CSL-Daily[3] | CSL | 21K | 2K | 10 | Lab | Fixed | 1 | Fixed | ALL |
| How2Sign[1] | ASL | 35K | 16K | 11 | Lab | Fixed | 1 | Fixed | ALL |
| DGS Corpus[8] | DGS | 63K | 23k | 327 | Lab | Fixed | 1 | Fixed | ALL |
| OpenASL[13] | ASL | 99k | 33k | ∼200 | Web | Diverse | 1 | Fixed | PARTIAL |
| BOBSL[2] | BSL | 1.2M | 7.8k | 39 | TV | Diverse | 1 | Fixed | PARTIAL |
| Auslan Corpus[19] | Auslan | - | - | 300 | Lab | Fixed | 1 | Fixed | ALL |
| **Auslan-Daily(ours)** | **Auslan** | **25K** | **14K** | **67** | **TV&Web** | **Diverse** | **1-10** | **Diverse** | **ALL** |

online Auslan corpora, including natural disaster reports and interviews (Figure 1, right). We collect these high-quality video materials, which capture various environments and include diverse daily life topics, namely Auslan-Daily.

To better translate sign videos in Auslan-Daily, we devise a two-stage data annotation labelling process, i.e., (1) aligning video clips and transcriptions and (2) detecting the signer in each aligned video clip. The rationale behind these two labelling stages is as follows: in (1), despite the original whole sign videos accompanying English subtitles, misalignments often exist between each sign video clip and its corresponding complete subtitle; In (2), explicitly labelling the signer's position helps simplify multi-person scenes and reduce translation complexity. In the first stage, we invite five Auslan experts to perform multi-gained annotations, including the temporal boundaries of fingerspellings, long-tailed isolated glosses, and sentences. In the second stage, we apply Alphapose [16, 17, 18], a tool for keypoints estimation and person tracking, to obtain each person's unique ID and pose sequences in each sign video clip. Then, for each sign clip, we manually modify the error of the Alphapose results and labelled the signer. After 300 working hours, we complete the fine-grained annotation labelling processing for our collected more than 45 hours of high-quality Auslan videos. Finally, in Auslan-Daily, there are around 2k video ↔ fingerspelling, 3k video ↔ gloss, and 25k video ↔ sentence pairs[7].

Based on these multi-grained annotations, we are able to investigate various sign language-related tasks in multi-person scenarios, including sign language translation, signer detection, fingerspelling detection, sign spotting, isolated sign language recognition and alignment. Specifically, we apply publicly available state-of-the-art models for each task and then report the performance with corresponding evaluation metrics. Experimental results demonstrate the challenges of Auslan-Daily due to its rich diversity and complexity. Overall, the contributions of this work are threefold:

- We construct the first large-scale Auslan dataset, dubbed Auslan-Daily, which contains multi-person sign videos on various topics in diverse environments.

- Auslan-Daily provides multi-grained annotations, enabling researchers to investigate a variety of tasks, including signer detection, fingerspelling detection, sign spotting, isolated sign language recognition, sign language alignment, and sign language translation.

- We establish a leaderboard and an evaluation benchmark to promote Auslan SLT research.

## 2 Related Work

### 2.1 Sign Language Translation Datasets

Since current models for sign language translation are significantly data-driven and based on deep learning, assembling extensive sign language datasets becomes a pivotal component for advancing sign language translation across various nations. Datasets constructed for SLT in recent years are shown in Table 1, where DGS, KSL, CSL, ASL, BSL, and Auslan represent German, Korean, Chinese, American, British, and Australian Sign Languages, respectively. Due to the limited lexical variety and sentence complexity within the SIGNUM [10] and KETI [9] datasets, they are unsuitable for SLT tasks. PHOENIX-2014T [4] is the first sign translation dataset used to translate German Sign Language (DGS). However, since this dataset only consists of weather forecast videos, the topic coverage is limited and may need to be increased for daily communication.

---

[7]All the gloss and fingerspelling video clips can be found in sign sentence video clips. While fingerspelling and gloss videos have some overlaps, fingerspelling videos are not the subset of gloss annotations.

BOBSL [2, 20] and OpenASL [13] are large-scale sign language translation datasets. For these two extensive datasets, experts verification has been confined solely to the validation and test sets, whereas the training set has been annotated via a pre-trained BSL sign alignment model [21] or through "self-generated" time boundaries based on ASL News. How2Sign [1] is the largest American Sign Language (ASL) dataset captured in the lab environment. It contains a variety of annotations, including multi-view information, depth information, pose, and speech. CSL-Daily [3] and DGS Corpus [8] are the extensive datasets of Chinese Sign Language (CSL) and German Sign Language (DGS), respectively. How2Sign, CSL-Daily and DGS Corpus cover diverse topics. The datasets pertinent to Auslan are represented by Auslan Signbank [22] and Auslan Corpus [23, 19, 24]. The Auslan Signbank constitutes a dictionary with approximately 5,500 Auslan glosses, while the Auslan Corpus [23, 19, 24] is primarily confined to exploratory research by linguists and is not entirely accessible to the public. More importantly, the above datasets only consider one person (signer) in each sign video clip, and the signer appears at a controlled certain position in each sign video clip. However, in the real world, multiple people may perform sign gestures in a scene, and their positions are diverse. Detecting the actual signer and accurately translating their gestural signs within complex environments presents a novel challenge. Factors such as perspective, illumination and the presence of crowds introduce noise and complexity when translating sign language to spoken language. In contrast, we propose Auslan-Daily, the first publicly available real-world Auslan translation dataset. Meanwhile, we enrich the diversity and complexity by collecting source various data.

## 2.2 Tasks Associated with Sign Language

Currently, there are several tasks for investigating various granularities of sign language, such as fingerspelling (character), gloss (word) and sentence. All the sign language-related tasks aim to foster better sign language understanding and sign language translation development. The sign language-related task, including: (i) **Sign Language Translation (SLT)** [4] task is to translate a sign video clip to the corresponding spoken language. The current SLT models can be divided into three categories, Sign2Gloss2Text, Sign2(Gloss+Text), and Sign2Text [25]. The Sign2Gloss2Text [26] model is a two-stage method, Sign2Gloss and Gloss2Text, respectively. The first stage is sign language recognition, which predicts a gloss sequence from a video, and the second stage translates the predicted gloss into the target natural language. Both Sign2(Gloss+Text) [27, 3, 28, 29, 30] and Sign2Text [31, 32, 33, 34] are end-to-end models. The difference is that Sign2(Gloss+Text) jointly trains the sign language recognition and translation tasks, and uses the gloss information as auxiliary supervision to extract features from videos, thereby improving the translation results. Though Sign2(Gloss+Text) models perform well on existing datasets, obtaining large-scale sign language translation data with continuous glosses annotation is costly and time-consuming. The primary reason is that the process of annotating one hour of continuous sign language video [21] requires an expert, who is proficient in sign language, approximately ten to fifteen hours. Thus, it is hard to apply them to SL datasets that do not contain gloss annotations. For Sign2Text models, they aim to directly convert sign language performed by a single person into target natural language. Furthermore, as the existing datasets are not collected in the wild, previous models might fail to tackle complex scenarios and real-world situations with multiple people; (ii) **Sign Language Alignment** [21, 35] temporally aligns asynchronous subtitles in sign language videos. A proficient alignment model for sign language can mine more sign data for automated translation; (iii) **Isolated Sign Language Recognition** [36, 37, 20] focuses on identifying and understanding individual gestural signs, independent of any surrounding context or sequence of signs; (iv) **Sign Spotting** [38, 39, 40] aims to find accurate locations of the given isolated signs in continuous co-articulated sign language videos; (v) **Fingerspelling Detection** [41, 42, 43] finds the fingerspelling segments' intervals within the clip. Fingerspelling is an important component of Sign Language, in which words are signed letter by letter and (vi) **Active Signer Detection**, also known as Sign Language Detection [44, 45], is identical to the initial stage of Signer Diarisation [46]. It aims to find the signer in the sign video clip.

## 3 Auslan-Daily Dataset

In this section, we describe data collection and cleaning, detail the data labelling procedure and provide statistics of the Auslan-Daily train/test split.

Table 2: Key statistics of Auslan-Daily. Auslan-Daily Communication and Auslan-Daily News are two sub-datasets split from Auslan-Daily. OOV: out-of-vocabulary. Singleton: words that only occur once in the training dataset.

| Sub-Dataset | Auslan-Daily Communication | | | Auslan-Daily News | | | |
|---|---|---|---|---|---|---|---|
| Domain/Topic | Communication | | | News & Documentary | | | |
| Video Resolution@FPS | 1920×1080@25 | | | 1280×720/1920×1080@29.97 | | | |
| Split | Train | Dev | Test | Train | Dev | Test | Total |
| Segments | 12,441 | 800 | 800 | 9,665 | 700 | 700 | 25,106 |
| Signers | 49 | 12 | 9 | 18 | 17 | 17 | 67 |
| Frames | 930,321 | 45,369 | 45,171 | 2,072,475 | 144,819 | 142,893 | 3,381,048 |
| Vocab. | 3,064 | 522 | 469 | 12,346 | 2,872 | 2,885 | 13,945 |
| Tot. words | 88,167 | 4,126 | 4,115 | 163,268 | 11,376 | 11,530 | 282,582 |
| Tot. OOVs | - | 8 | 10 | - | 326 | 304 | - |
| Singletons | 1,043 | - | - | 5,267 | - | - | - |
| Person per clip | 1-11 | 1-8 | 1-8 | 1-8 | 1-8 | 1-7 | 1-10 |

## 3.1 Data Collection and Cleaning

"Sally and Possum", "ABC News with Auslan" and Auslan corpora from YouTube are public TV programs and open sources[8]. "Sally and Possum", a premium Australian sign language show for deaf children. To facilitate children learning of Auslan, the production team designs the plots and writes the English scripts. Subsequently, Auslan experts perform sign language based on the script. This show has 6 seasons and 15 episodes per season, which is 22.5 hours long and contains around 20 topics, including daily communication, study skills, and knowledge explanation. Beginning in 2022, "ABC News with Auslan" weekly broadcasts key domestic and international events news and weather forecasts. It is an ongoing TV program, and the current dataset includes 45 news videos as of May 2023. Auslan experts execute a real-time sign language translation (simultaneous interpretation) for deaf or people with hearing loss based on currently broadcast news. However, the English subtitles may leave the following problems [21]: (i) the order of subtitles is not complying between spoken and sign languages, and (ii) the duration of a subtitle varies considerably between signing and speech due to differences in speed and grammar. To enrich the topics of our dataset, we also collect several publicly available high-quality documentaries interpreted with Auslan. These data comprise specific thematic areas, including disaster overviews, preventative measures, and interviews.

All of the original videos are recorded with standard English dubbing and subtitles. We download subtitles of each whole original video, which are arranged with the format "[Start Time] subtitle [End Time]". The time intervals are marked based on dubbing. Upon spot check, we observe that longer subtitles might extend across multiple temporal intervals, whereas several shorter subtitles tend to appear within a time interval. To procure complete sentence-level subtitles, we conduct the data cleaning operations as follows: (1) for incomplete subtitles, *e.g.*, ending with a comma, we connect them with the following subtitles to compose complete sentences and merge their time duration; (2) for several complete subtitles that appear within a time interval, we partition them into multiple independent sentences; (3) for a complete sentence that only contains modal particles, *e.g.*, "Oh!" and "Ha ha ha!", we remove them to avoid meaningless translation. As a result, we acquire approximately 29k complete subtitles that require alignment.

## 3.2 Two-Stage Data Labelling Procedure

To obtain applicable data for sign language translation, we design two stages of the data labelling procedure: (1) aligning video clips and transcriptions and (2) detecting the signer in each aligned video clip. They are imperative as original subtitles often misalign with sign videos, and the position of the signer varies. By analyzing the frequency of individual tokens across all subtitles, we notice a considerable presence of long-tailed words. Due to their sparse occurrence, these words impose challenges for the model and affect the translation results. To investigate this problem, sign language experts annotate the less commonly used glosses in "Sally and Possum". Experts also label the temporal boundaries of significant fingerspelling instances, as fingerspelling is commonly used in the deaf community. Therefore, in the first stage, we engage Auslan experts to synchronise video ↔ fingerspelling, video ↔ gloss, and video ↔ sentence pairs.

Next, we employ Alphapose [16, 17, 18] to track people in each aligned video clip. Then, we record trajectories and pose sequences along with their corresponding IDs. Considering Alphapose

---

[8]Our dataset follows the copyright **Creative Commons BY-NC-ND 4.0** license ©.

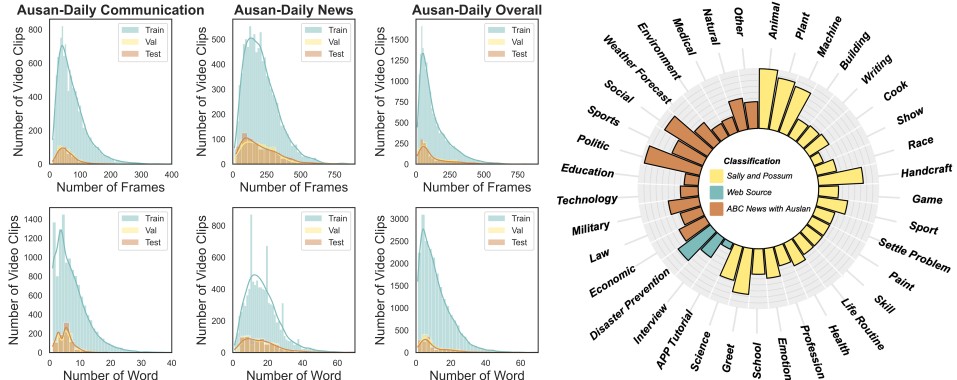

Figure 3: Distributions of #frames/#words over clips (L) and the topic diversity in Auslan-Daily (R).

may suffer tracking errors sometimes[9], we thus invite annotators to manually check and modify the tracking results by assigning correct IDs to the signers.

To guarantee the annotation quality of our dataset, we conduct a cross-check verification process during each data labelling procedure stage. Specifically, we ask each Auslan annotator as an examiner to cross-check around 5% of annotated video clips provided by another annotator. The video clips are chosen randomly. If the examiner finds more than 10% of annotated videos have obvious errors, a third annotator is invited to review and correct the annotations.

Through the collaborative efforts of five Auslan experts and five annotators, we complete all annotations with approximately 300 work hours. Overall, our Auslan-Daily dataset contains the following **annotations**: (1) temporal boundaries of sign video clips; (2) temporal boundaries of long-tailed glosses; (3) temporal boundaries of partial fingerspellings; (4) pose sequences of signers and non-signers; (5) bounding-boxes of signers; (6) signer identities and (7) English transcriptions. These multi-grained annotations can be further investigated for Australian sign language-related tasks.

### 3.3 Data Statistics

During the data labelling, experts discard erroneous data, such as subtitles do not have corresponding sign language. As demonstrated in Table 2, there are in total 25,106 video clips encompassing 67 unique signers, with the vocabulary size of 13,945 words. It should be noted that signers in the validation and test sets appear in the training set. As the number of persons ranges from 1 to 10 in a video clip, the distractions, such as gesture interference of multi-persons, are also involved in sign language translation, thus imposing challenges in this task. To verify the robustness of the various models, we also statistics of the sentences within the test set. As shown in Table 3 (Appendix), over 80% of video clips in the test set encompass distinct sentences. Therefore, the robustness of the models can be verified by evaluating on the test set.

After examining the domains of the three data sources, "ABC News with Auslan" and the data from online Auslan corpora are found significantly similar. Thus, we combine the data collected from these two video sources and partition the Auslan-Daily dataset into two sub-datasets: Auslan-Daily Communication and Auslan-Daily News. We randomly split the two sub-datasets data into the training, validation and test sets as shown in Table 2. Each sign video clip has 73 frames and 7.8 words on average for the Auslan-Daily Communication sub-dataset. Each sign video clip has 214 frames and 16.9 words on average for the Auslan-Daily News sub-dataset. The distributions of frames and words in different splits of the sub-datasets are shown in Figure 3 (left).

Moreover, Auslan-Daily has 3,000 (600 classes) long-tailed isolated glosses and 2,000 significant fingerspellings. For the isolated gloss, we select words appearing less than ten times after the natural language corpus post-lemmatization[10] in "Sally and Possum". Its main purpose is to enhance translation models to recognise long-tail words through sign language recognition or spotting tasks. Furthermore, considering the pivotal role of fingerspelling in sign language translation, we thus

---

[9]The overall error rate is less than 20 percent of video clips with multiple signers.

[10]Lemmatization facilitates a more fine-grained and accurate analysis and representation of text.

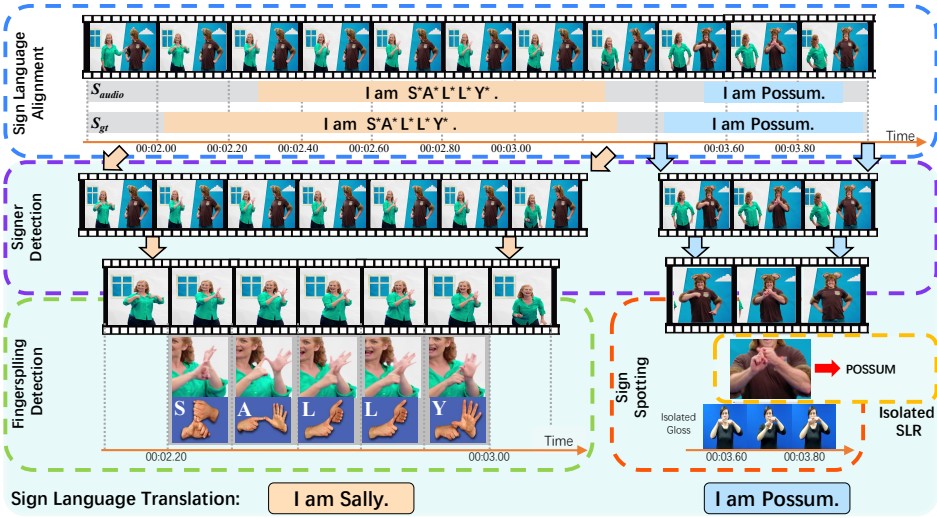

Figure 4: Overview of Auslan-Daily tasks and provided annotations.

annotate Auslan fingerspelling instances. As illustrated in Figure 3 (right), there are 21 types of daily communication and more than 15 types of daily news in our Auslan-Daily dataset.

# 4 Overview of Auslan-Daily Tasks

## 4.1 Task Definition

**Active Signer Detection** (ASD) [44, 45, 46]: Given trajectories of $N$ people $\mathcal{P} = \{p_i\}_1^N$ in a sign video clip $\mathbb{V} = \{f_i\}_1^t$, the goal of ASD is to find the active signer $p_i$ in the current video clip.

**Sign Spotting** (SS) [38, 39]: Given a sign video clip $\mathbb{V}$ with $t$ frames and an isolated gloss $\mathbb{I}$ clip, the goal of SS is to locate the start frame $f_m$ and end frame $f_n$ of $I$ in $\mathbb{V}$.

**Isolated Sign Language Recognition** (ISLR) [36, 37, 20]: Given an isolated gloss $\mathbb{I}$ clip, ISLR aims to determine the gloss category $c$ of $\mathbb{I}$, where $c$ is pre-defined by a Sign Dictionary.

**Fingerspelling Detection** (FD) [41]: Given a sign video clip $\mathbb{V}$ with $t$ frames, the goal of FD is to locate the start frame $f_m$ and end frame $f_n$ of each fingerspelling in $\mathbb{V}$.

**Sign Language Alignment** (SLA) [21]: Given a whole original video $\mathcal{V} = \{f_i\}_1^T$ with $T$ frames and its corresponding M subtitles $\mathcal{S} = \{s_i\}_1^M$, the goal of SLA is to align each subtitle sentence $s_i$ with the precise begin frame $f_i$ and end frame $f_j$.

**Sign Language Translation** (SLT) [4]: Given a sign video clip $\mathbb{V}$ with $t$ frames, the goal of SLT is to translate $\mathbb{V}$ to an English sentence $s_i$.

## 4.2 Evaluation Metrics

**BLEU and ROUGE:** BLEU [47] and ROUGE [48] scores are commonly-used evaluation metrics for *Sign Language Translation* [4]. BLEU-n measures the n-gram overlap between the generated text and reference text, and ROUGE-L measures the F1 score based on the longest common subsequences between the generated text and reference text.

**Top-$K$ Accuracy:** Top-$K$ classification accuracy measures the number of ground-truth labels within the top $k$ predicted labels. For *Signer Detection*, $K = 1$ is employed. For *Isolated Sign Language Recognition*, we adopt $K$ values of 1, 5, and 10.

**IoU:** Intersection over Union (IoU) is defined as the ratio of the intersection and the union of the predicted and actual time intervals of an action within a video [49, 50]. Similar to [41], we employ AP@IOU as a metric for *Fingerspelling Detection*. Following [21, 39], we use F@IOU to evaluate *Sign Language Alignment* and *Sign Spotting* tasks.

Table 3: Translation results of Single/Multi-Person SLT gloss-free models on Auslan-Daily.

| Single-Per. SLT | Input | Auslan-Daily Communication | | | | | Auslan-Daily News | | | | |
|---|---|---|---|---|---|---|---|---|---|---|---|
| | | R | B1 | B2 | B3 | B4 | R | B1 | B2 | B3 | B4 |
| SL-Luong [59] | **Pose** | **37.27** | **30.15** | **16.26** | 11.67 | 9.45 | 20.65 | **19.84** | **7.81** | **4.59** | **2.81** |
| SL-Luong [59] | RGB | 13.49 | 13.54 | 7.85 | 5.74 | 4.66 | 16.14 | 16.92 | 7.44 | 4.07 | 2.68 |
| SL-Transf [27] | Pose | 35.65 | 31.31 | 16.17 | 11.41 | 9.20 | 20.25 | 21.25 | 6.57 | 3.32 | 2.11 |
| SL-Transf [27] | RGB | 14.97 | 15.25 | 9.05 | 6.51 | 5.20 | 14.93 | 17.64 | 7.41 | 3.98 | 2.52 |
| TSPNet-Joint [31] | RGB | 26.89 | 26.07 | 10.07 | 5.46 | 3.76 | 19.71 | 18.23 | 5.97 | 3.21 | 2.26 |
| MMTLB [29] | RGB | 17.64 | 18.35 | 14.27 | 9.76 | 6.11 | 18.90 | 19.64 | 5.30 | 3.26 | 2.31 |
| GASLT [33] | Pose | 35.74 | 28.19 | 16.00 | **11.93** | **9.95** | 18.76 | 15.57 | 6.06 | 3.72 | 2.72 |
| GASLT [33] | RGB | 31.46 | 25.05 | 10.18 | 6.25 | 4.73 | **22.01** | 19.54 | 7.45 | 4.41 | 2.56 |
| **Multi-Per. SLT** | **Input** | R | B1 | B2 | B3 | B4 | R | B1 | B2 | B3 | B4 |
| SL-Luong [59] | RGB | 14.21 | 13.58 | 6.63 | 3.83 | 2.45 | 14.04 | 15.53 | 6.11 | 3.27 | 2.05 |
| SL-Transf [27] | RGB | 13.53 | 14.45 | 7.58 | 4.48 | 2.86 | 13.68 | 16.58 | 5.86 | 2.72 | 1.55 |
| TSPNet-Joint [31] | RGB | 27.86 | 26.38 | 10.08 | 5.00 | 3.28 | 14.64 | 17.33 | 3.86 | 1.66 | 1.89 |
| MMTLB [29] | RGB | 20.53 | 16.54 | 11.32 | 6.84 | 4.52 | 17.76 | 16.02 | 4.81 | 2.83 | 1.83 |
| GASLT [33] | RGB | 29.33 | 23.62 | 9.44 | 5.69 | 4.23 | 19.73 | 16.99 | 6.25 | 3.44 | 2.26 |
| **SD+SLT** | **Pose** | **34.28** | **28.94** | **14.90** | **10.49** | **8.38** | **19.43** | **17.22** | **7.12** | **4.13** | **2.53** |

# 5 Auslan-Daily Benchmark

## 5.1 Video Representation

**Pose-based video feature representation:** Pose-based representations are robust against background clutters, lighting conditions, and occlusions, while explicitly depicting human hand and limb movements [51, 52, 53]. Several recent studies exploit pose information and achieve state-of-the-art performance in sign language translation-related tasks [9, 54, 55, 36, 56]. Hence, we use the key points extracted from Alphapose[16, 17, 18] as video features to provide benchmark results.

**RGB-based video feature representation:** Several models directly extract features from sign videos, such as CNN-RNN-HMM network [4], S3D [57], and I3D [58]. In the works [20, 36, 31], I3D is used for sign video representation. To better adapt to SL dataset and capture the spatio-temporal information of signs, inspired by [31], we finetune I3D on a word-level sign language recognition dataset and extract sign video features with different window widths and strides.

## 5.2 Benchmark Results

In this section, we provide benchmark results of sign language translation, alignment, active signer detection, fingerspelling detection, sign spotting and isolated sign recognition tasks on Auslan-Daily.

**Sign Language Translation:** In this task, we employ publicly available gloss-free SLT models, including (1) the RNN-based language translation model (SL-Luong [59]), (2) the Temporal Semantic Pyramid network (TSPNet [31]), (3) the Sign Language Translation Transformer model without glosses (SL-Transf [27]), (4) the multi-modality transfer learning based model (MMTLB [29]) and (5) transformer-based model with the gloss attention mechanism (GASLT [33]). Note that these models are designed to translate sign videos that only contain one single signer. To meet the input requirement of these SLT models, we crop the signer regions based on the ground-truth bounding-boxes of the signers. Since Auslan-Daily videos are captured in diverse scenes, RGB-based representation models may be affected by background clutter and various camera angles. In our single-person SLT experiments, GASLT performs best on the Auslan-Daily Communication subset, while SL-Luong excels on the Auslan-Daily News subset, both using pose points (hands & body) as input. In addition, to shed some light on how existing SLT models perform on Auslan-Daily with multiple persons in each video clip (without cropping acting signers), we directly feed video clips into SLT models. As indicated by Table 3, we observe significant performance degradation. This implies that existing SLT models do not have attention mechanisms to focus on active signers and non-signers can distract SLT. Therefore, we introduce a paradigm of signer detection followed by a translation model, denoted by SD+SLT, in Table 3. It is observed that SD significantly facilitates translation in multi-person scenarios.

**Sign Language Alignment:** We use Subtitle Aligner Transformer (SAT) [21] model to evaluate the sign language alignment task. It employs Transformer [60] to synchronise subtitles with BSL videos and provides a pre-trained model [2]. The results are shown in Table 4. Leveraging the pre-trained

Table 4: The baseline of Sign Language Alignment (SLA) on Auslan-Daily. *Comm.*, *News* and *Mixed* refer to two sub-datasets and the total combined dataset, respectively.

| Fine-Tune | Test | F1@.10 | F1@.25 | F1@.50 |
|---|---|---|---|---|
| **News** | News | 80.59 | 72.30 | 50.08 |
| **Comm.** | Comm. | 87.58 | 77.41 | 66.33 |
| **Mixed** | Comm. | 81.28 | 71.81 | 64.27 |
| | News | 85.13 | 77.00 | 53.05 |
| | Mixed | 82.49 | 73.43 | 60.76 |

Table 5: The baseline of Signer Detection (SD) and Isolated Sign Language Recognition (ISLR) on Auslan-Daily. *B* and *Hs* represent Body and Hands keypoints, respectively.

| Task | Model + Feature | Top-1 | Top-5 | Top-10 |
|---|---|---|---|---|
| **SD** | TGCN + B + HS | 88.35 | - | - |
| | I3D + Video | **89.01** | - | - |
| **ISLR** | TGCN + B + HS | **14.82** | **25.37** | **37.32** |
| | I3D + Video | 11.87 | 20.10 | 30.33 |

model provided by [21] and fine-tuning the sign language alignment model on Auslan enhances alignment performance.

**Active Signer Detection:** Active Signal Detection can be considered a binary action recognition problem. Inflated 3D ConvNet (I3D) model [58] is employed as the baseline on the RGB-based model. In addition, Pose-based Temporal Graph Convolution Networks (Pose-TGCN) [36, 61] serve as the pose-based model baseline. As shown in Table 5, Pose-TGCN and I3D achieve similar performance.

**Isolated Sign Language Recognition:** Isolated Sign Recognition is a multi-class action classification task. Similar to ASD, I3D and Pose-TGCN are adopted for the RGB-based and pose-based, respectively. Table 5 illustrates that the pose-based model performs better in complex scenes.

**Fingerspelling Detection:** We evaluate Auslan-Daily fingerspelling detection using the publicly available state-of-the-art fingerspelling detection model [41]. It is a multi-task model that combines pose estimation, recognition and detection tasks to improve the detection results jointly. The performance for the fingerspelling detector achieves 0.33/0.28/0.21 for AP@IoU(0.1/0.3/0.5). Comparing fingerspelling in Auslan-Daily Communication with Auslan-Daily News reveals more complexity and faster speed of fingerspelling in the latter.

**Sign Spotting:** We employ the sign spotting model [39] to evaluate the sign spotting task on the Auslan-Daily sign potting task. This method fusion multi-modal features, including RGB and poses, to obtain sign clip representations. Meanwhile, it introduces an innovative top-k transferring technique during testing to reduce the domain gap between isolated signs and continuous sign language. The baseline of this task is an F1 score of 0.27.

## 6 Discussion and Limitation

**Reasons for Low Performance on Auslan-Daily News Translation:** (1) Table 2 indicates that the dictionary size of Auslan-Daily News is much larger than that of Auslan-Daily Communication and Auslan-Daily News is long-tailed. These impose challenges in translating Auslan from videos to English [12, 2, 13]. (2) The experts commonly use abbreviations for named entities in videos, such as person and organization names. However, the corresponding words in the subtitles are in the full form. (3) Since "ABC News with Auslan" is a real-time TV program, the experts may summarise and interpret the simultaneous broadcasting news to Auslan, which could lead to missing words. (4) Fingerspelling recognition can be significantly affected by different camera angles and signing speeds, and thus current models struggle to identify each word, especially in news.

**Reasons for Low Performance on Long-Tailed Isolated Sign Language Recognition:** Since we annotate isolated signs which are distributed in the long tail in the vocabulary, the average number of sign videos for each sign is much smaller than that of normal ISLR dataset [36, 37]. Moreover, as our sign videos include diverse signers, environments and camera perspectives, our ISLR videos are more challenging. This scenario is also practical in SLT tasks since not every sign has a large number of corresponding videos. As a result, the ISLR accuracy in Table 5 is low than existing benchmarking results. As demonstrated in Table 5 (Appendix), we also adopt more methods to evaluate ISLR.

**Coreference Resolution:** Coreference occurs when two or more expressions refer to the same person or thing [62]. It is observed that translating Auslan faces the challenge of coreference resolution [15, 63]. Signers frequently employ coreference to nouns to reduce the signing complexity. Consequently, we suggest exploiting context information to address coreference resolution in sign language translation [64, 65, 66, 67].

**Cross-Domain Investigation:** The Auslan-Daily dataset is subdivided into Auslan-Daily Communication and Auslan-Daily News. The differences in sources and topics naturally create a domain gap. Auslan-Daily provides a practical dataset for investigating the cross-domain issue inherent in the sign language translation task [68, 69, 70, 55].

**The Volume of Datasets:** Our project aims at an ongoing exploration of machine sign language translation for Auslan. Unlike ASL and BSL, the high-quality data corpora of Auslan are relatively small. Therefore, we will continue incorporating content from "ABC News with Auslan" and other Auslan corpora to enrich Auslan-Daily. We intend to leverage existing Auslan data to provide additional annotations, similar to BOBSL [2]. In other words, we can employ a trained Auslan alignment model for preliminary annotations [21] and then conduct manual verification by experts.

# 7   Conclusion

In this work, we propose the first pubic available large-scale multi-person Auslan translation dataset with multi-grained annotation, named Auslan-Daily. Moreover, Auslan-Daily includes diverse topics and multiple signers performing in various environments. More importantly, the collected sign conversations are captured in the wild, significantly increasing the challenges of Auslan translation. Extensive experiments demonstrate the validity and challenges of our Auslan-Daily. Thanks to the multi-grained annotations, our dataset can be used for other sign language-related tasks. Furthermore, the presented benchmark results can act as strong baselines for future research. Although Auslan-Daily currently only has English transcriptions, we intend to provide gloss annotations and the benchmark on Continuous Sign Language Recognition (CSLR) in the future to further promote research on Auslan translation.

## Acknowledgement

This research is funded in part by ARC-Discovery grant (DP220100800 to XY), ARC-DECRA grant (DE230100477 to XY) and Google Research Scholar Program. We gratefully thank all the anonymous reviewers and ACs for their constructive comments.

## Broader Impact

Auslan, like many other sign languages, has its distinctive features in semantics and pragmatics. The complexity and diversity of the expressions of Auslan present a significant hurdle in designing vision-language models. In this paper, the challenges posed by Auslan-Daily help stimulate the development of the vision-language community.

Moreover, as a visual language that unfolds in a three-dimensional space, sign language poses perspective-related complexities, such as capturing signs from a side-view perspective, that differ from written and spoken languages. Incorporating sign language into deep learning networks entails addressing these specific challenges. Releasing this dataset is part of open science. This dataset can encourage and support other researchers to conduct research on Auslan. By utilising this dataset, researchers can develop and improve algorithms and applications that understand and generate Auslan. For example, real-time sign language translation systems can be developed, making it easier for deaf individuals to communicate with others and enhancing their social participation and quality of life. This will also contribute to increased awareness and understanding of the Australian deaf community and foster broader social engagement.

Apart from computer science, our dataset will provide new opportunities for interdisciplinary research. For instance, linguists can use it to study the linguistic features and variations of Auslan, social scientists can investigate the culture and social interactions of the Australian deaf community, and psychologists and neuroscientists can explore the cognitive and neural mechanisms underlying sign language processing and learning. More broadly, the deaf community and sign language users are often overlooked in dataset and technological advancements. This oversight can lead to biases and imbalances within AI models as well. Our releasing this dataset will help bridge that gap and provide necessary data resources for creating more equitable and inclusive AI systems. In the rapidly advancing era of AI, it is of paramount importance to ensure that the needs and inclusion of the deaf community are not overlooked.

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
