# OpenReview forum: "Auslan-Daily: Australian Sign Language Translation for Daily Communication and News"
_NeurIPS.cc/2023/Track/Datasets_and_Benchmarks — NeurIPS 2023 Datasets and Benchmarks Poster_

### Official Review · Reviewer_qbZ3 · 2023-07-20
**Comments for paper 47**

**Rating:** 7
**Confidence:** 3
**Correctness:** I think the proposed dataset is const…
**Clarity:** The paper writing is easy to follow.

**Strengths:**

The proposed dataset is with large-scale, fully annotated, and can be used for various tasks.

From the experimental results, it can be seen that Auslan-Daily is a highly challenging SLT dataset.
This makes the proposed dataset promising to contribute to the development of Auslan, even the advancement of sign languages worldwide in a broader context.

All datasets and benchmarks have been available to the public.

**Additional Feedback:**

N/A

**Documentation:**

This paper provides a URL for reviewer access to the dataset.

**Ethics:**

I have no other ethical concerns with the submission.

**Limitations:**

 The authors adequately addressed the limitations of their work in Section 6 Discussion and Limitation.

**Opportunities For Improvement:**

 The proposed dataset is a large-scale Auslan dataset. For the advancement of worldwide sign language translation, the comparisons between this dataset and the state-of-the-art widely-used SL datasets in other languages, e.g., American English, and Chinese, are encouraged to be clarified.

Are there any similar datasets (including the scale, annotations, and tasks) in other languages? What is the difference and advantage of the proposed one besides it is in Auslan?

**Relation To Prior Work:**

This paper has discussed how this work differs from previous works, especially the Auslan dataset.

**Summary And Contributions:**

In this work, the authors construct a large-scale Auslan dataset Auslan-Daily, which contains multi-person sign videos with various topics and diverse environments.

Auslan-Daily provides multi-grained annotationsfor multiple tasks, including signer detection, fingerspelling detection, sign spotting, isolated sign language recognition, sign language alignment, and sign language translation.

This work also establishes a leaderboard and an evaluation benchmark to promote Auslan SLT research.

---

> ### Author Response · Authors · 2023-08-22
> **Response Q1-Q2**
>
> We sincerely thank the reviewer **qbZ3** for the valuable comments and suggestions. We are glad to see that the reviewer acknowledges Auslan-Daily is a challenging SLT dataset and contributes to the development of Auslan translation. We respond to the questions raised by the reviewer below.
>
> **Q1: Comparison between Auslan-Daily and widely-used sign language datasets in other languages.**
>
> **R1:** In Table 1 (main paper), we compare our Auslan-Daily with other widely-used sign language datasets from different countries, such as PHOENIX-2014T [4] from German, CSL-Daily [3] from China, How2Sign [1] from USA, and BOBSL [2] from UK. As indicated in Table 1 (main paper), in terms of corpus scales, Auslan-Daily is moderate, but its annotations are all manually labelled and checked, while the other larger-scale sign datasets are partially annotated. In terms of topic and scene diversity, Auslan-Daily not only contains diverse topics but also involves more various scenes, including multiple signer scenarios, different camera angles and diverse scene setups, in comparison to other sign language datasets. The visual comparisons between Auslan-Daily and widely-used sign language datasets are illustrated in Figure 2 (main paper). We emphasised these differences in the revision.
>
> **Q2: Are there any similar datasets (including the scale, annotations, and tasks) in other languages? What is the difference and advantage of the proposed one besides it is in Auslan?**
>
> **R2:** Thank you for the comments. There are similar datasets in other sign languages in terms of scale, such as CSL-Daily [3] and How2Sign [1]. They are captured in the lab environments, while our Auslan-Daily is collected from real-world TV shows and News. It involves various camera angles, illumination conditions, conversation scenes and signers. Though some datasets also include in-the-wild scenarios, such as BOBSL [2] and OpenASL [13], their data are partially annotated. In contrast, all the videos in Auslan-Daily are manually labelled.
>
> In comparison to other datasets frequently used in sign language-related tasks, Auslan-Daily has the following advantages:
> 1. Auslan-Daily contains multi-grained annotations. These allow researchers to conduct various sign language-related tasks, such as alignment, sign spotting, and fingerspelling detection, on Auslan-Daily.
> 2. Auslan-Daily involves real-world Auslan conversation and News broadcasting scenes. The corpus in Auslan-Daily is much closer to the daily usage of Auslan, thus facilitating the deployment of sign language models in practice.
> 3. Auslan-Daily not only includes a wide range of topics but also contains multiple signers in a scene. These impose more challenges to current SLT tasks and thus push the frontier of SLT in the wild.

---

### Official Review · Reviewer_vHjw · 2023-07-20
**Auslan-Daily: Australian Sign Language Translation for Daily Communication and News**

**Rating:** 8
**Confidence:** 5
**Clarity:** It is generally well written.

**Strengths:**

In general, the paper is well written. The design of the dataset is well thought. The major strengths of the paper are:
1. obviously a new relatively large dataset for Auslan as there are not many large SL datasets even in other languages. Here, the dataset is "large" in terms of number of signers which 67 and vocabulary which is 14K.
2. it covers both in-the-wild scenarios and more predictable scenarios such as news report, weather forecast and documentaries with possibly diverse backgrounds. Few current SLT datasets contain in-the-wild scenarios.
3. it covers situations that there can be more than 1 signer in a scene, and thus SLT will require signer tracking and perhaps requires dealing with signer interference. Again, few current SLT datasets contain multi-signer situations.
4. multi-grained annotations are provided at the sentence level for the whole dataset, and at the gloss level and fingerspelling for some subsets of the dataset.
5. train/dev/test data splits are given and benchmark evaluation results are given using publicly available recent SLT models so that future research on the new dataset can be properly gauged.

**Additional Feedback:**

N/A

**Correctness:**

I believe most parts are correctly performed. My only minor reservations are
1. if the annotation procedure includes any verification step
2. the benchmark results are weak. Of course, that may really be just due to the in-the-wild scenarios.

**Documentation:**

Good.

**Ethics:**

Fine.

**Limitations:**

Not really.

**Opportunities For Improvement:**

Again the paper is well written in general. The following comments are provided to improve the clarity of the paper in some aspects.
1. The paper and Table 2 say that there are a total of 67 signers. But from Table 2, it seems 67 signers only count those in the training sets unless the signers in the dev/test sets also appear in the training sets. The authors need to clarify this.
2. Annotation is the most human-intensive step in the labeling process. In any case, is there any verification protocol? As annotators may make mistakes, is each annotation checked by the same annotator or another annotator for a number of times?
3. As the authors also mention, the SLT and ISLR performance are really low with BLEU-4 in a single digit (9.45 in single signer scenario) and ISLR top-1 accuracy of 14.8%. OpenASL has a vocabulary of 33K and ~200 signers and only gets a BLEU-4 of 6.72. I will expect that this dataset will get "much" better results with fewer signers and smaller vocabulary. Note I am talking about the the low BLEU-4 score in single signer situation, so the weak results are not due to multiple signers.
4. The conclusion mentions providing gloss labels in the future. May the author elaborate more on how to do that as it will require a lot of manpower? It is not easy at all.
5. A minor typo: line #216: should be Figure 3 not Figure 4.

**Relation To Prior Work:**

yes.

**Summary And Contributions:**

The paper presents a new sign language (SL) dataset called Auslan-Daily, for the Australian SL or Auslan mainly for SL translation (SLT), but it will also support other tasks such as signer detection (SD), sign spotting (SS), etc. It consists of 45 hours of video and is claimed to be the largest for Auslan SLT. It is composed of two subsets. The first is Auslan educational TV series involving daily communications among multiple signers in the wild, and the second is Auslan TV programs on news, weather forecast and documentaries. The topics are diverse, and there are multiple signers in a scenario which is captured by multiple cameras at various positions. The data are annotated by experts at various level of details, e.g.,  at the sentence level, gloss level and fingerspelling through a 2-stage labelling procedure. Benchmark results are given using recent SLT models.

The major contributions are:
1. the creation of a large scale Australian sign language data set
2. multi-grained annotations.
3. evaluation benchmark is given.

---

> ### Author Response · Authors · 2023-08-22
> **Response Q1-Q5**
>
> We sincerely thank the reviewer **vHjw** for the constructive comments and suggestions. We are appreciated that the reviewer recognises that our key contributions on the creation of a large-scale Auslan dataset, multi-grained annotations and evaluation benchmarks. We address the concerns raised by the reviewer below.
>
> **Q1: The counting signers in the training and dev/test sets.**
>
> **R1:** Thank you for the comments. Since we randomly split all the video clips into training and dev/test sets, signers in the validation and test sets appear in the training set. We clarified this in the revision.
>
> **Q2: The verification protocol in labelling process.**
>
> **R2:** To guarantee the annotation quality of our dataset, we conduct a cross-check verification process. Specifically, we ask each Auslan annotator as an examiner to cross-check around 5% of annotated video clips provided by another annotator. The video clips are chosen randomly. If the examiner finds more than 10% of annotated videos have obvious errors, a third annotator is invited to review and correct the annotations. We added the discussion in the revision.
>
> **Q3: The reason for the low performance in SLT and ISLR in Auslan-Daily compared with OpenASL.**
>
> **R3:** Thank you for the comments. We follow the official training protocols of all publicly available SLT models and then test their models on Auslan-Daily in Table 3 (main paper). Moreover, we also include a newly-released work GASLT [32] (CVPR23). GASLT achieves **9.95** and **2.72** in BLEU-4 on our communication and news subset, respectively. The comparable performance of existing SLT methods not only indicates the performance boundary of current SLT models but also reflects the challenges of our datasets. We believe that our evaluation is appropriate.
>
> **Table 1. Performance of single-person sign language translation based on GASLT [32] on Auslan-Daily.**
> | Model        | Input       | Comm. ROUGE  | Comm. BLEU-4 | News ROUGE  | News BLEU-4 |
> |--------------|:-------------:|:----------:|:----:|:----------:|:----:|
> | GASLT [32]  | Pose |  **35.74**  |  **9.95**  |  18.76  | **2.72**  |
> | GASLT [32]  | RGB  |  31.46  |  4.73  |  **22.01**  | 2.56 |
>
> Although OpenASL has more signers and larger vocabulary, it simplifies the test subtitles. On the contrary, we did not simplify test subtitles while removing too simple and repeated test samples in our test set. Thus, our testing set is more challenging than OpenASL. That is the reason we did not see obvious performance improvement on SLT compared to OpenASL.
>
> Furthermore, since we annotated isolated signs which are distributed in the long tail in the vocabulary, the average number of sign videos for each sign is much smaller than that of normal ISLR dataset [35,36]. Thus, the benchmarking results in [35,36] are not comparable to ours. Moreover, as our sign videos include diverse signers, environments and camera perspectives, our ISLR videos are more challenging. As a result, the ISLR accuracy in Table 5 (main paper) is lower than existing benchmarking results. We clarified this in the revision.
>
> **Q4: Plan for continuous gloss annotations.**
>
> **R4:** Annotating continuous glosses is indeed a time-consuming and laborious task.
> Therefore, we will adopt an iterative human in-the-loop procedure to annotate Auslan videos. To be specific, we plan to annotate videos as follows:
> 1. Auslan experts will annotate continuous glosses for commonly used sign sentences (around 10-20% of the entire data).
> 2. With the annotated corpus, we will train a continuous sign language recognition (CSLR) model.
> 3. We then apply our trained CSLR model to predict continuous glosses for unseen data. Afterwards, Auslan experts will need to modify the incorrect pseudo labels of continuous glosses rather than manually align every gloss with corresponding frames.
> 4. We repeat the step 2 and step 3 to achieve a more accurate CSLR model that can provide better semi-automatic annotations.
>
> By employing this progressive labelling procedure, high-quality continuous gloss annotations can be achieved more efficiently and at scale.
>
> **Q5: A minor typo in Figure 3 (main paper).**
>
> **R5:** Thank you for pointing it out. We have corrected the typo in the revision.

---

> > ### Comment · Reviewer_vHjw · 2023-08-26
> > **Satisfactory clarification**
> >
> > Thanks for the clarification and addressing my questions.

---

### Official Review · Reviewer_4ztv · 2023-07-21
**Well written and obvious contributions**

**Rating:** 6
**Confidence:** 3
**Clarity:** This paper is well written.

**Strengths:**

Overall, the paper is well written and understandable. The dataset is well annotated and usable for multiple tasks.

**Additional Feedback:**

In this paper, the authors propose a Auslan-daily dataset that collects more than 45 hours of high-quality Auslan video materials and provide multi-grained annotations for different sign language tasks. This dataset fills a gap in the lack of dedicated large-scale sign language datasets in Australia. Overall, the paper is well written and understandable. The dataset is well annotated and usable for multiple tasks. According to the quality of the paper, my comments are listed as follows:

1.	It appears that there have been previous Auslan datasets, for example, Johnston, T. (2010). From archive to corpus: Transcription and annotation in the creation of signed language corpora. International Journal of Corpus Linguistics, 15(1), 106-131. The authors should consider making a comparison between their dataset and these previous ones.
2.	In section 5.2, most tasks are tested with a small number of methods. I suggest the authors add more methods for evaluation to demonstrate the challenges of the proposed dataset.
3.	Do the test sets of the proposed dataset contain many cases that are not seen in the training set to verify the robustness of various models?


**Correctness:**

As a benchmark, it constructs the dataset reliably and provides three evaluation metrics.

**Documentation:**

The composition of this dataset is reasonable, but it needs to explain whether the test set contains more challenging cases. In addition, this dataset provides a project page, which visually shows several cases, the introduction from partial to overall and the composition of the data set, etc.

**Limitations:**

In the section 6 of this paper, the authors discuss some special experimental phenomena and the dataset's limitations from four points, which I think is sufficient.

**Opportunities For Improvement:**

1.	It appears that there have been previous Auslan datasets, for example, Johnston, T. (2010). From archive to corpus: Transcription and annotation in the creation of signed language corpora. International Journal of Corpus Linguistics, 15(1), 106-131. The authors should consider making a comparison between their dataset and these previous ones.
2.	In section 5.2, most tasks are tested with a small number of methods. I suggest the authors add more methods for evaluation to demonstrate the challenges of the proposed dataset.
3.	Do the test sets of the proposed dataset contain many cases that are not seen in the training set to verify the robustness of various models?


**Relation To Prior Work:**

It appears that there have been previous Auslan datasets, for example, Johnston, T. (2010). From archive to corpus: Transcription and annotation in the creation of signed language corpora. International Journal of Corpus Linguistics, 15(1), 106-131. The authors should consider making a comparison between their dataset and these previous ones.

**Summary And Contributions:**

In this paper, the authors propose a Auslan-daily dataset, which collects more than 45 hours of high-quality Auslan video materials and provide multi-grained annotations for different sign language tasks. This dataset fills a gap in the lack of dedicated large-scale sign language datasets in Australia. Overall, the paper is well written and understandable. The dataset is well annotated and usable for multiple tasks.

---

> ### Author Response · Authors · 2023-08-22
> **Response Q1-Q2**
>
> We sincerely thank the reviewer **4ztv** for the valuable comments. We appreciate the reviewer for recognising that our proposed dataset fills a gap in the lack of dedicated large-scale sign language datasets in Australia. We address the concerns raised by the reviewer below.
>
> **Q1: Comparison between Auslan Corpus (Johnston, T., 2010) and Auslan-Daily.**
>
> **R1:** Thank you for the comment. As mentioned on Auslan Corpus's official [website](https://www.elararchive.org/dk0001/), its data include personal opinions on sensitive subjects and personal information. Due to the ethical concerns, their data are not publicly available. In contrast, Auslan-Daily is the first public large-scale Auslan dataset, which contains multi-person sign videos on various topics in diverse environments. Compared with Auslan Corpus that is captured under controlled environments, Auslan-Daily also imposes challenges to sign language translation in the wild scenarios.
>
> **Q2: More methods for evaluation to demonstrate the challenges of the proposed dataset.**
>
> **R2:** Thank you for the valuable suggestion. As suggested, we further incorporate more methods to evaluate the performance of other sign language-related tasks on our Auslan-Daily. To be specific, we adopt TSN [A], SlowFast [B] and Timesformer [C] to perform isolated sign language recognition and signer detection, S_audio [D], S_audio+ [D] (whether shifted audio is employed for alignment) and Segment SLV [E] to perform sign language alignment, Bi-LSTM CTC [F] and Modified R-C3D [G] to perform fingerspelling detection, and HS-I3D [H] and Two-Stage-SP [I] to perform sign spotting. Thanks to the advanced networks, we achieved better performance. These results also shed some light on how to further improve the sign language-related tasks in the future. We included these new results and discussion in the revision.
>
> **Table 1. The baseline of Isolated Sign Language Recognition on Auslan-Daily.**
> | Model | Top-1 |  Top-5 |
> | -------- | :--------: | :--------: |
> |  TSN [A] |   24.75    |  **41.31**   |
> |  SlowFast [B] |  23.97     |  38.25   |
> |  Timesformer [C] |   **27.38**    |  40.33   |
>
> **Table 2. The baseline of Signer Detection on Auslan-Daily.**
> | Model | Top-1  |
> | -------- | :--------: |
> |  TSN [A] |   89.98    |
> |  SlowFast [B] |   90.37    |
> |  Timesformer [C] |   **93.65**    |
>
> **Table 3. The baseline of Sign Language Alignment on Auslan-Daily.**
> | Model | F1@.10 |  F1@.25 | F1@.50 |
> | -------- | :--------: | :--------: | :--------: |
> |  S_audio [D] |  50.68 | 31.54 | 22.30 |
> |  S_audio+ [D] |  74.87 | 66.32 | 29.65 |
> | Segment SLV [E] |   77.43    |  69.49   |  41.76  |
> | SAT [21] | **82.49** | **73.43** | **60.76** |
>
> **Table 4. The baseline of Fingerspelling Detection on Auslan-Daily.**
> | Model | AP@IoU 0.1 |  AP@IoU 0.3 | AP@IoU 0.5 |
> | -------- | :--------: | :--------: | :--------: |
> |  Bi-LSTM CTC [F]  |    0.25   |   0.12   |  0.09  |
> |  Modified R-C3D [G]  |   0.30    |   0.24   |  0.18  |
> |  Multi-FD [40]  |    **0.33** | **0.28** | **0.21** |
>
> **Table 5. The baseline of Sign Spotting on Auslan-Daily.**
> | Model | F1 score |
> | -------- | :--------: |
> |  HS-I3D [H]  |   **0.35**    |
> |  Two-Stage-SP [I]  |   0.23  |
> |  Dual-Branch SP [38] | 0.27  |
>
> [A] Wang, L, et al., Temporal segment networks: Towards good practices for deep action recognition. ECCV 2016.
>
> [B] Feichtenhofer, C, et al., Slowfast networks for video recognition. ICCV 2019.
>
> [C] Bertasius, G, et al., Is Space-Time Attention All You Need for Video Understanding? ICML 2021.
>
> [D] Bull H, et al., Aligning Subtitles in Sign Language Videos. ICCV 2021.
>
> [E] Bull H, et al., Automatic segmentation of sign language into subtitle-units. ECCVW 2020.
>
> [F] Huang, X, et al., Bidirectional LSTM-CRF Models for Sequence Tagging. ArXiv Preprint 2015.
>
> [G] Xu, H, et al., R-C3D: Region convolutional 3D network for temporal activity detection. ICCV 2017
>
> [H] Wong, R, et al., Hierarchical I3D for Sign Spotting. ECCVW 2022.
>
> [I] Chen, X, et al., The solution for the Sign Spotting Challenge at ECCV. ECCVW 2022.

---

> > ### Author Response · Authors · 2023-08-22
> > **Response Q3**
> >
> > **Q3: Do the test sets of the proposed dataset contain many cases that are not seen in the training set to verify the robustness of various models?**
> >
> > **R3:** Thank you for the comment. We confirm that each video clip in our test sets has not been seen in the training set. We provide the statistics of the sentences in the test set in Table 6. Even though there might be similar English subtitles, every sign video clip in the test sets is still different from those in the training sets. This is because those sign sentences are signed by different signers, captured under different backgrounds or under different camera perspectives. Hence, we can guarantee that test samples are not included in the training set.
> > Moreover, over 80% of video clips in the test set present unique sentences. In other words, these sentences never appear in the training set. Therefore, the robustness of models can be verified by evaluating on the test set. We clarified this in the revision.
> >
> > **Table 6. Statistics of unseen words and sentences in the test set.**
> > |  |  Auslan-Daily Communication |  Auslan-Daily News |
> > | -------- | :--------: | :--------: |
> > | Num. Sentence in test set   | 800      | 700     |
> > | Num. Unseen Words       | 10       | 304     |
> > | Num. Unseen Sentences   | 564      | 662     |

---

### Official Review · Reviewer_MEP1 · 2023-07-21
**Reviews of MEP1**

**Rating:** 8
**Confidence:** 5
**Clarity:** The paper is well written.

**Strengths:**

1. It is the first large-scale Australian sign language dataset, which is not only helpful to the research on Auslan but also to the community of sign language understanding.
2. The vocabulary size is large, and the background and topics are diverse.
3. A high resolution of 1080p.
4. The dataset can support multiple sign language understanding tasks, e.g., sign language translation, sign spotting, and isolated sign recognition.
5. The dataset consists of conversational sign videos, which are deficient in existing sign language datasets.

**Additional Feedback:**

A typo in Table1: persons per clip.

**Correctness:**

The claims are correct. But some benchmark results need to be explained more as stated above.

**Documentation:**

The dataset has a good documentation. A URL pointed to the dataset is available.

**Ethics:**

No ethical concerns.

**Limitations:**

The authors have adequately discussed limitations.

**Opportunities For Improvement:**

1. One key task, continuous sign language recognition (CSLR), is not mentioned in the paper. The authors need to explain why CSLR is not included although the gloss annotations are available.
2. The benchmark results need more explanations. First, the performance is pretty low in Table 3. Are the models listed gloss-free or gloss-based? Besides, Table 3 only shows results involving one single modality. However, many sign language works, e.g., TwoStream-SLT [R1], propose to jointly model both RGB videos and keypoints, and a clear improvement can be seen over the single-modality models. It will be better to include dual-modality baselines.
3. The ISLR accuracy in Table 5 is also very low. Does the model training converge?
4. In line 72, the dataset consists of different types of annotations. What is the relationship among the videos belonging to different annotation types. For example, are the videos (2k) with fingerspelling annotations a subset of those (3k) with gloss annotations?
5. The reason of using AlphaPose. It seems that most previous works use HRNet to extract keypoints. Are there any advantages of using AlphaPose?

[R1] Two-Stream Network for Sign Language Recognition and Translation, NeurIPS 2022.

**Relation To Prior Work:**

Yes. A detailed comparison can be seen in Table 1.

**Summary And Contributions:**

The paper introduces a large-scale Australian Sign Language (Auslan) dataset, Auslan-Daily. It is collected from two sources including educational TV series and Auslan Daily news. Both the topics and backgrounds of the dataset are diverse, and vocabulary size is quite large (~14K). Moreover, benchmark results are provided for various sign language understanding tasks, e.g., sign language translation, sign spotting, and isolated sign language recognition. The dataset can definitely enrich the resources for sign language understanding, which will be a highly challenging benchmark.

---

> ### Author Response · Authors · 2023-08-22
> **Response Q1-Q5**
>
> We sincerely thank the reviewer **MEP1** for the appreciative and constructive comments. We sincerely appreciate that the reviewer recognises our paper's strengths, including the Auslan-Daily's unique contributions to the sign language understanding field. We address the concerns raised by the reviewer below.
>
> **Q1: The authors need to explain why Continuous Sign Language Recognition (CSLR) is not included.**
>
> **R1:** Thank you for the comment. In this work, we only annotate isolated long-tailed glosses for Isolated Sign Language Recognition (ISLR). Since labelling continuous glosses for each sign sentence is very costly and time-consuming, these annotations are not available yet. We are currently dedicated to annotating continuous glosses. Once we complete continuous gloss annotations, we will release the annotations to the community. Then, CSLR can be conducted in our dataset in the future. We clarified this in the revision.
>
> **Q2: The authors should explain the low performance in Table 3 (main paper) and whether the models listed are gloss-free or gloss-based?**
>
> **R2:** Thank you for the comment. The performance reported in Table 3 (main paper) is from **gloss-free SLT models**. The performance of gloss-free SLT models in general is lower than that of gloss-based models due to the lack of continuous gloss supervision. Moreover, our Auslan-Daily is collected from real-world TV shows and News, and thus it involves various camera perspectives, illumination conditions and conversation scenes. These factors increase the difficulty of SLT in the wild. Moreover, Auslan-Daily has a 50% larger vocabulary size than PHOENIX-2014T [4] and CSL-Daily [3], and the state-of-the-art method GASLT [32] only achieves 15.74 (BLEU-4) on PHOENIX-2014T and 4.07 (BLEU-4) on CSL-Daily when continuous gloss annotations are not available. We believe once continuous gloss annotations are available, we can further improve SLT performance of Auslan.
>
> **Q3: It will be better to include dual-modality baselines.**
>
> **R3:** Thank you for the constructive suggestion. As suggested, we include the dual-modality baseline TwoStream-SLT [A] in our benchmark. To be specific, we feed both video frames and keypoints into the network. Note that TwoStream-SLT was originally trained with continuous gloss supervision, and thus its original performance is higher than its gloss-free version. The gloss-free version of TwoStream-SLT achieves **7.69** and **2.41** in BLEU-4 on the communication and news subset, respectively. Compared with RGB-based representations, pose-based representations are more robust against background clutters, lighting conditions, and occlusions. Here, we observed that dual-modality features are affected by the RGB-branch stream, and thus the result is slightly lower on TwoStream-SLT [A]. We included this discussion in the revision.
>
> [A] Chen, Y., et al., Two-stream network for sign language recognition and translation. NeurIPS 2022.
>
> **Q4: The ISLR accuracy in Table 5 (main paper) is also very low. Does the model training converge?**
>
> **R4:** Since we annotated isolated signs which are distributed in the long tail in the vocabulary, the average number of sign videos for each sign is much smaller than that of normal ISLR dataset [35,36]. Moreover, as our sign videos include diverse signers, environments and camera perspectives, our ISLR videos are more challenging. This scenario is also practical in SLT tasks since not every sign has a large number of corresponding videos. As a result, the ISLR accuracy in Table 5 (main paper) is low.
> We evaluate the ISLR performance on the test set when the performance of models has not improved on the validation set after two consecutive epochs. Therefore, we believe our models have converged. We clarified the discussion in our revision.
>
> **Q5: What is the relationship among the videos belonging to different annotation types? For example, are the videos (2k) with fingerspelling annotations a subset of those (3k) with gloss annotations?**
>
> **R5:** We annotate video clips including **fingerspelling**, **isolated glosses**, and **sign sentences**. In our benchmark, all the gloss and fingerspelling video clips can be found in sign sentence video clips. While fingerspelling and gloss videos have some overlaps, fingerspelling videos are not the subset of gloss annotations. For instance, many words are only signed by fingerspelling once or twice. Thus we do not include them in gloss annotations as they are not sufficient samples for training and testing, but we include them in fingerspelling video clips since fingerspelling detection focuses on the start and end time when fingerspelling occurs rather than what fingerspelling refers to. We clarified the relationship among the video clips in the revision.

---

> > ### Author Response · Authors · 2023-08-22
> > **Response Q6-Q7**
> >
> > **Q6: Reason for using AlphaPose instead of HRNet for keypoint extraction.**
> >
> > **R6:** AlphaPose is the latest and most accurate multi-person pose estimator. It not only provides keypoint extraction but also integrates the pose-tracking function. In Auslan-Daily, most of videos contain multiple signers. Thus, pose-tracking in AlphaPose significantly facilitates us to localise acting and non-acting signers in multi-person scenarios.
> > We thank the reviewer for the suggestion. As suggested, we evaluate the SLT performance based on the poses extracted by HRNet, and the results are shown in Table 1. We observe that the SLT performance based on HRNet is lower than that using AlphaPose because AlphaPose achieves higher and more reliable keypoint estimation results, especially hands [B]. Furthermore, considering the benefit of the pose-tracking functionality integrated in AlphaPose, we opt to choose AlphaPose in our work. We added the discussion in the revision.
> >
> > **Table 1. Performance of single-person sign language translation on Auslan-Daily Communication (Comm.) and News .**
> > | Model        | Input       | Comm. ROUGE  | Comm. BLEU-4 | News ROUGE  | News BLEU-4 |
> > |--------------|:-------------:|:----------:|:----:|:----------:|:----:|
> > | SL-Luong [58]          | HRNet |  34.86  |  8.04  |  **21.57**  | 2.15  |
> > | SL-Transf [26]  | HRNet |  36.23  |  8.82  |  19.43  | 2.00  |
> > | SL-Luong [58]  | AlphaPose |  **37.27**  | **9.45**  |  20.65  | **2.81**  |
> > | SL-Transf [26]  | AlphaPose |  35.65  |  9.20  |  20.25  | 2.11  |
> >
> > [B] Fang, H., et al., AlphaPose: Whole-Body Regional Multi-Person Pose Estimation and Tracking in Real-Time. TPAMI 2022.
> >
> > **Q7: A typo in Table 1 (main paper).**
> >
> > **R7:** Thanks for pointing out the typo. We have corrected this typo in the revision.

---

> > > ### Comment · Reviewer_MEP1 · 2023-08-26
> > >
> > > Thanks for the authors' rebuttal. My concerns are well addressed. I recommend the authors to include more details and baseline results in the revised version. I believe the dataset would be impactful in the sign language understanding community. I will increase my rating to 8.

---

### Official Review · Reviewer_fC6g · 2023-08-01
**A nice dataset on Australian Sign Language**

**Rating:** 6
**Confidence:** 4
**Correctness:** The paper seems to be technically cor…

**Strengths:**

S1. The dataset is large and consists of significant variation among the subjects covered.

S2. The dataset has multiple levels of annotations, including fingerspellings. This is a great initiative.

S3. The total number of signers covered is also decently large, which adds to the variation present in the dataset.

S4. The dataset collection strategy and cleaning strategies are presented well.

**Additional Feedback:**

None

**Clarity:**

The authors are clear about their contributions and the description of the dataset.

**Documentation:**

The authors have provided documentation of the dataset as well as a link to the dataset. They have also provided the baselines in their GitHub repository.

**Ethics:**

No major concerns

**Limitations:**

W1. The authors should try to present better baselines where the performance on the dataset is more accurate. Currently, the scores in the case of SLT are pretty low, and thus it leaves some doubt about whether the dataset is correctly cleaned or not.

W2. The authors can present some baseline results on Sign Language Production too on their dataset.

W3. The authors can present some ablation studies and try out different representations of the visual stream to get a better understanding of their dataset.

**Opportunities For Improvement:**

Check the limitations below.

**Relation To Prior Work:**

This is the first large Australian Sign language dataset.

**Summary And Contributions:**

The paper proposes a new sign language dataset on Australian Sign Language. The dataset has multiple levels of annotations and is well described in the paper.

---

> ### Author Response · Authors · 2023-08-22
> **Response Q1-Q2**
>
> We sincerely thank the Reviewer **fC6g** for the constructive comments. We are glad that the reviewer acknowledges Aulsan-Daily is a large, diverse Australian Sign Language dataset with multi-level annotations. We address the concerns raised by the reviewer below.
>
> **Q1: The authors should try to present better baselines. It leaves some doubt about whether the dataset is correctly cleaned or not.**
>
> **R1:** Thank you for the suggestion. We have tried our best to evaluate all publicly available Sign Language Translation (SLT) models in Table 3 (main paper). We further evaluate a recently published SLT model GASLT [A]. As shown in Table 1, GASLT achieves **9.95** and **2.72** in BLEU-4 on our communication and news subsets, respectively.
>
> **Table 1. The baseline of Single-Person Sign Language Translation (SLT) gloss-free models on Auslan-Daily.**
> | Model        | Input       | Comm. ROUGE  | Comm. BLEU-4 | News ROUGE  | News BLEU-4 |
> |--------------|:-------------:|:----------:|:----:|:----------:|:----:|
> | GASLT [32]  | Pose |  **35.74**  |  **9.95**  |  18.76  | **2.72**  |
> | GASLT [32]  | RGB  |  31.46  |  4.73  |  **22.01**  | 2.56 |
>
> Auslan-Daily is collected from real-world TV shows and News, and thus it involves various camera perspectives, illumination conditions, conversation scenes and signers. These factors increase the difficulty of SLT in the wild. Auslan-Daily has a larger vocabulary size than PHOENIX-2014T [4] and CSL-Daily [3]. Note that GASLT only achieves 15.74 (BLEU-4) on PHOENIX-2014T and 4.07 (BLEU-4) on CSL-Daily. Moreover, unlike OpenASL, we do not simplify the testing subtitles and also remove too simple yet repeated sentences. Thus, our testing set is more challenging than OpenASL. The low performance of SLT should not be attributed to the quality of the annotations.
>
> Furthermore, we ensure the annotation quality of our dataset through a cross-check verification process. Specifically, we ask each Auslan annotator as an examiner to cross-check around 5% of randomly-chosen video clips labelled by another annotator. If more than 10% of annotated videos have obvious errors, a third annotator will review and correct the annotations. We added this explanation in the revision.
>
> [A] Yin, A., et al., Gloss Attention for Gloss-free Sign Language Translation. CVPR 2023.
>
> **Q2: The authors can present some baseline results on Sign Language Production on their dataset.**
>
> **R2:** Thank you for the comment. To evaluate Sign Language Production (SLP), accurate 3D keypoints of signers are often required. However, when we apply state-of-the-art 3D pose estimation methods to our dataset, they all fail to provide precise 3D poses, especially for hand gestures. This is because the resolution of hand areas might not be as high as the datasets specific for 3D hand pose estimation. Moreover, in our Auslan-Daily, signers may not face cameras in a frontal view and there are self-occlusions and occlusions by other objects. These factors impose difficulties in accurate 3D pose estimation. Employing generative adversarial networks for SLP is another option. However, we found the cluttered background in Auslan-Daily significantly impedes network learning and the hand gestures are barely recognisable in the generated videos.
>
> To provide a baseline for SLP, we replace the 3D keypoints with 2D keypoints in Text2Pose (T2P) [B]. As indicated in Table 2, T2P achieves 0.61 and 0.54 in BLEU-4 on the communication and news subsets, respectively. Without precise 3D keypoints, the diverse orientation of signers inevitably introduces ambiguity to SLP. Additionally, compared to PHOENIX-2014T [4], the larger vocabulary size of Auslan-Daily further imposes challenges on SLP. Therefore, we reckon the current annotations of Auslan-Daily (missing accurate 3D poses) may not be sufficient for high-quality SLP. In the future, we will consider annotations of 3D poses. In that case, SLP can be accurately evaluated on Auslan-Daily. We clarified this discussion in the revision.
>
> **Table 2. Sign Language Production (SLP) performance of Text2Pose [B] on Auslan-Daily.**
> | Sub-dataset  | ROUGE  | BLEU-1 | BLEU-2  | BLEU-3 | BLEU-4
> |--------------|:----------:|:----:|:----------:|:----:|:----:|
> | Communication  |  16.30  |  24.54  |  5.86  | 1.58  | 0.61 |
> | News           |  26.29  |  16.85  |  4.45  | 1.53  | 0.54 |
>
> [B] Saunders, B., et al., Progressive transformers for end-to-end sign language production. ECCV 2020.

---

> > ### Author Response · Authors · 2023-08-22
> > **Response Q3**
> >
> > **Q3: Ablation studies with different visual stream representations.**
> >
> > **R3:** Thank you for the valuable suggestion. To dissect the impacts of different visual representations, we evaluate sign language translation methods with different network architectures (RNN-based and Transformer-based models), window sizes and pre-trained backbones.
> >
> >
> > **Table 3. Ablation study on different visual representations of Single-Person SLT on Auslan-Daily Communication.**
> > | Model | Pre-training Dataset      | Window Size | ROUGE  | BLEU-4 |
> > |--------------|:----------:|:----:|:---------:|:---------:|
> > | SL-Luong [58]| WLASL [35]    | 8/12/**16** |  13.49  |  4.66  |
> > | SL-Transf [26] | WLASL [35]  | 8/12/**16** |  14.97  |  5.20  |
> > | **SL-Luong [58]** | **MSASL** [36]   | 8/**12**/16 |  **19.10**  |  **6.94**  |
> > | SL-Transf [26]| MSASL [36]   | 8/**12**/16 |  16.58  |  4.90  |
> > | SL-Luong [58] | BSL [20]     | 8/12/**16** |  12.55  |  3.54  |
> > | SL-Transf [26] | BSL [20]     | 8/12/**16** |  15.98  |  4.01  |
> > | SL-Luong [58]  | BSL+WLASL [35]| 8/**12**/16 |  14.71  |  5.23  |
> > | SL-Transf [26] | BSL+WLASL [35]| **8**/12/16 |  14.18  |  4.41  |
> > | SL-Luong [58]  | BSL+MSASL [36]| 8/**12**/16 |  15.21  |  6.43  |
> > | SL-Transf [26] | BSL+MSASL [36]| 8/**12**/16 |  14.56  |  4.57  |
> >
> >
> > **Table 4. Ablation study on different visual representations of Single-Person SLT on Auslan-Daily News.**
> >
> > | Model | Pre-training Dataset      | Window Size | ROUGE  | BLEU-4 |
> > |--------------|:----------:|:----:|:---------:|:---------:|
> > | **SL-Luong [58]**  |  **WLASL** [35] | 8/12/**16** |  **16.14**  |  **2.68**  |
> > | SL-Transf [26] |  WLASL [35] | 8/12/**16** |  14.93  |  2.52  |
> > | SL-Luong [58]   | MSASL [36]   | 8/12/**16** |  16.68  |  2.31  |
> > | SL-Transf [26]  | MSASL [36]   | 8/**12**/16 |  15.43  |  2.45  |
> > | SL-Luong [58]   | BSL [20]     | 8/**12**/16 |  12.95  |  1.48  |
> > | SL-Transf [26]  | BSL [20]     | **8**/12/16 |  11.07  |  1.42  |
> > | SL-Luong [58]   | BSL [20] + WLASL [35]| 8/**12**/16 | 14.82   | 1.97   |
> > | SL-Transf [26]  | BSL [20] + WLASL [35]| **8**/12/16 |  15.70  |  2.08  |
> > | SL-Luong [58]   | BSL [20] + MSASL [36]| 8/12/**16** |  15.76  |  2.33  |
> > | SL-Transf [26]  | BSL [20] + MSASL [36]| 8/**12**/16 |  14.29  |  1.95  |
> >
> > As shown in Table 3, using MSASL [36] and the window size of 12, the model performs the best on Auslan-Daily Communication while using WLASL [35] and a window size of 16 as shown in Table 4, the model performs the best on Auslan-Daily News. These experiments indicate the differences between the communication and News corpora, and challenges inherent in sign language translation in the wild.

---

### Decision · Program_Chairs · 2023-09-22

**Decision:**

Accept (Poster)

**Comment:**

All the reviewers express positive opinions on the paper, which creates a new sign language dataset on Australian Sign Language with benchmark results provided. The dataset is well annotated (by experts at various level of details) and usable for multiple tasks. Also one leaderboard is established for promoting Auslan SLT research.
Therefore, the paper should be accepted.